# Digital health literacy is linked to attitudes regarding the ethical aspects of digital health among patients with dermatologic comorbidities

Ana Lilia Ruelas-Villavicencio[1,2], Irazú Contreras-Yáñez[1,3], Roxana Paola Gómez-Ruiz[2], María Clara Zagaglia Del Valle[2], Andrea Malagón-Liceaga[4], Virginia Pascual-Ramos[1,3]*

1 Centro Interdisciplinario de Bioética de la Universidad Panamericana (CIBUP), Mexico, 2 Department of Dermatology, Instituto Nacional de Ciencias Médicas y Nutrición Salvador Zubirán, Mexico, 3 Department of Immunology and Rheumatology, Instituto Nacional de Ciencias Médicas y Nutrición Salvador Zubirán, Mexico, 4 Physiology Department, School of Medicine, Universidad Nacional Autónoma de México, Mexico

* virtichu@gmail.com

## Abstract

### Introduction

Digital health literacy (DHL), also known as eHealth literacy, refers to an individual's ability to locate, understand, evaluate, and apply health information from electronic sources to make informed health decisions. This skill is increasingly regarded as essential for navigating the modern healthcare landscape, promoting health equity, and improving health outcomes. The study objective was to establish an association between DHL and dermatologic outpatients' attitudes regarding ethical aspects of digital health. Additionally, we validated a questionnaire designed to assess these bioethical attitudes.

### Patients and methods

This cross-sectional study was performed in two phases (April 2024-December 2024). Phase-1 consisted of validating the Bioethical Attitudes toward Digital Health questionnaire (BADH). Phase-2 evaluated the association between the eHEALS (it assesses a person's ability to use digital health resources) and BADH scores. Three convenience samples of consecutive patients were used: S-1 included 46 patients who participated in a pilot testing, S-2 included 100 patients who participated in the BADH validation and S-3 included 120 patients and was used to investigate the association between DHL and bioethical attitudes. Descriptive statistics and multiple linear regression analysis were used.

**Data availability statement:** The study used outpatient data from a public center in Mexico City, derived from clinical records considered sensitive personal information under Mexican law. For confidentiality, individual data are not publicly available. Participants provided informed consent guaranteeing that their personal information will not be disclosed. Only aggregated results are shared. Access to the complete dataset requires a reasonable request and approval from the local Institutional Review Board (IRB). Data may be requested from the corresponding author or directly through the Research Ethics Committee (comite.etica. investigacion@incmnsz.mx).

**Funding:** The author(s) received no specific funding for this work.

**Competing interests:** The authors have declared that no competing interests exist.

## Results

The 8-item BADH was found to be feasible, valid, and reliable. The exploratory factor analysis revealed a two-factor structure, consisting of trust and privacy dimensions, which accounted for 59.8% of the total variance. This structure was subsequently validated through confirmatory factor analysis. The BADH reliability was confirmed with a Cronbach's alpha of 0.686 and ICC of 0.684 (95% CI: 0.581–0.770). A positive linear association was identified between the eHEALS and the BADH scores ($\beta = 0.465$, 95%CI: 0.218–0.450, $p < 0.001$). This relationship was evident with the trust dimension of the BADH ($\beta = 0.526$, 95%CI: 0.206–0.379, $p < 0.001$), but not with the privacy dimension.

## Conclusions

DHL is associated with individual moral positions regarding digital health, particularly those concerning trust. The BADH questionnaire has adequate psychometric properties.

## Introduction

Digital technologies for managing health have rapidly expanded in recent years, largely due to the COVID-19 pandemic [1]. These include telehealth, patient portals, and mobile health applications. Telehealth facilitates the delivery of healthcare services remotely using information and communication technology, which has been particularly useful for ensuring individuals receive treatment when in-person visits aren't possible [2]. Clinical trials support the efficacy of telehealth in enhancing clinical outcomes across various conditions [3]. Furthermore, it may be more cost-effective than other methods, and patients generally express high satisfaction with their experiences [4].

Although access to technology has improved over the past ten years, significant disparities remain for those with lower incomes, limited education, or from racial and ethnic minority groups [5]. This issue, referred to as the digital divide, affects health equity [6]. A key factor in addressing this divide is evaluating whether individuals not only have access to the internet and digital devices but also have the necessary skills to use them effectively for healthcare purposes. This concept is known as digital health literacy (DHL) [7,8]. A recent study of hospitalized patients revealed that many participants with low DHL had internet access and prior experience with digital devices but still required assistance to complete online tasks [9]. This finding suggests that DHL may pose a larger challenge to the effective use of digital resources than access itself, highlighting the need for a deeper understanding and measurement of the phenomenon [2].

Currently, twenty different assessment tools have been utilized to evaluate DHL, a recognized determinant of health [2,10–14]. Norman and Skinner [11] developed the eHEALS scale, the first instrument for assessing DHL, which is the most widely used

in the field. This scale has been applied to German patients with skin cancer [12] and validated in Spanish by Paramio Pérez et al. [13]. Its construct validation was also tested alongside a sociodemographic characterization survey [14].

Digital health ethics refers to the ethical considerations and principles that guide the development, deployment, and use of digital technology in healthcare [15]. These ethical aspects encompass a broad range of dilemmas. They include, but are not limited to, risks associated with health technology, user safety and well-being, security and privacy concerns, as well as issues related to transparency and accountability that may arise from utilizing digital health technologies [16–23].

Another important aspect of understanding the use of digital technologies extends beyond DHL and centers on individual attitudes. Attitude is a person's mental and emotional disposition toward something or someone, evident in their thoughts, feelings, and behaviors. It reflects an intention that drives specific actions. In sociology, attitude describes an individual's or group's tendency to respond to stimuli based on their values and beliefs [24]. The Moral Foundations Theory (MFT), developed by social psychologists, explains the broad range of human moral judgments, values, and behaviors through underlying moral intuitions and emotions [25]. This descriptive theory argues that moral judgment is primarily intuitive and non-rational, divided into discrete categories of moral intuition [26]. The MFT has been associated with individual attitudes toward health in Western democracies, with unintended consequences, including the promotion of ineffective—and potentially harmful—overreactions to authoritative recommendations [27,28].

Positive patient attitudes toward digital health tools in dermatology have been reported, highlighting their benefits. However, concerns regarding bioethical aspects have also been raised, including the potential loss of personal contact and changes in the patient-physician relationship. Additionally, doubts about privacy and security, as well as limited trust in these technologies, have been documented [29].

In summary, current research on DHL and technology access mainly focuses on North America, Europe, and China, leaving a significant gap regarding participants from Latin America (LATAM). The underrepresentation of certain demographics in digital health research limits the comprehensiveness of the topic [9]. DHL is increasingly recognized as a social determinant of health. Despite the growing attention on digital health tools, the ethical dimensions that may influence patient attitudes toward these technologies and their interaction with DHL remain understudied. These phenomena are particularly relevant for patients with dermatologic diseases, as both healthcare professionals and patients acknowledge the benefits of eHealth services in daily practice. Although the evidence supporting the effectiveness and cost-efficiency of these interventions is becoming more recognized, they continue to be underutilized [29].

Our hypothesis was that DHL may have a positive association with moral attitudes toward digital health. The primary study objective was to establish an association between DHL and patients' attitudes regarding ethical aspects of digital health in a dermatology service from an academic center located in Mexico City. We also validated a questionnaire to assess relevant bioethical attitudes regarding digital health (The BADH questionnaire. Please see the S1_Appendix) and explored the factors associated with DHL.

## Patients and methods

**Ethics.** On January 13, 2024, some patients from the dermatology service were invited to review the conceptualization and design of the study prior to its submission to the Institutional Review Board. The Research Ethics Committee of the Instituto Nacional de Ciencias Médicas y Nutrición Salvador Zubirán (INCMyN-SZ) approved the study on February 28, 2024 (Reference number: DER-4932-25-25-1). The first participant signed the informed consent form and entered the study on April 25, 2024. All participants provided written informed consent, and the study was conducted in accordance with the principles of the Helsinki Declaration [30].

## Study design and period

This cross-sectional study was conducted in two phases and included participants from April 25, 2024, to December 18, 2024.

Phase-1 consisted of developing and validating the BADH. It included at first questionnaire content validity, followed by pilot testing to confirm its face validity and feasibility, and then its construct validity and reliability.

Phase 2 evaluated the association between DHL and BADH scores in the target population. We also explored the factors associated with DHL. This phase involved the application of the previously validated BADH and the eHEALS, and the collection of relevant sociodemographic clinical and treatment information from consecutive patients attending the dermatology service of the INCMyN-SZ.

## Settings and study population

Patients were recruited from the dermatology outpatient clinic of the INCMyN-SZ, which provides care for approximately 5000 patients with comorbid dermatologic conditions.

During the study, consecutive adult outpatients who were able to provide informed consent and had a scheduled consultation at the dermatology clinic, with a confirmed diagnosis established by the attending dermatologist, were invited to participate (inclusion criteria). Patients were excluded if they had severe cognitive impairment that would hinder their ability to complete the questionnaires.

## Description of samples and sample size calculation

Three convenience samples of consecutive patients were used. Two samples were considered during phase-1: The first one (S-1) included 46 patients who participated in a pilot testing [31] and the second sample (S-2) included additional 100 patients [32,33]. Both samples were used for BADH validation. The third sample (S-3) included 120 patients and was used during phase-2, to establish the association between BADH and the eHEALS and to explore the factors associated with DHL. Non-probabilistic, intentional sampling was employed, including consecutive outpatients from the dermatologic clinic [34].

To achieve our primary objective, a sample size calculation was conducted using the appropriate formula for an F-test in the context of multiple linear regression. We hypothesized a moderate effect size for the association, represented by $f^2 = 0.15$, with a significance level (α) set at 0.05 and a power (1-β) of 0.80. The model included up to seven predictors, which comprised DHL as well as other significant variables and potential confounders such as sex, age, education level, and social media usage. As a result, a sample size of at least 103 participants was determined to be necessary.

To account for an anticipated 10% loss of non-analyzable data, we decided to increase the sample size to (at least) 113 patients.

## Patients evaluations

**Phase-1 procedures.** Expert judgement developed the BADH in accordance with FDA guidelines [35] and assessed its content validity. The BADH focuses on three key bioethics domains: data privacy (3 items), patient autonomy (2 items), and trust in digital health (3 items). A 5-point Likert scale was used to score the items: totally disagree, partially disagree, undecided, partially agree, and totally agree. Higher scores indicated a more favorable attitude towards the bioethical domains.

Ten experts independently evaluated the 8 items for relevance, appropriate wording, and clarity of meaning. They also assessed the face validity of the dimensions and the clarity of the instructions. These experts were selected based on their experience with dermatologic patients, knowledge of bioethics, digital health experience, reputation, availability, and impartiality [36]. A similar evaluation process was conducted with 46 patients from the target population during pilot testing to confirm content validity.

Construct validity was established through exploratory and confirmatory factor analysis [37].

Reliability was assessed by measuring internal consistency and temporal stability (test-retest), conducted with 51 patients who completed the questionnaire twice: at baseline and again one week later [37].

Finally, the feasibility of the BADH was evaluated among the 46 patients participating in the pilot testing. This assessment considered the time required to complete the questionnaire, acceptance of the format by patients, the number of incomplete questionnaires, and the overall time taken to fill out the questionnaire. At the end of the BADH feasibility, it was decided that patients might benefit from assistance when completing the questionnaire.

**Phase-2 procedures.** During this phase, the BADH was first applied, followed by 8-items eHEALS. Standardized formats were utilized to gather relevant sociodemographic information, clinical and treatment related information (Table 1). Data accuracy was ensured through a thorough review of medical charts and patient interviews.

## Instruments description

*eHEALS* **(8 items).** It is a questionnaire designed to assess a person's ability to use digital health resources. It measures how comfortable individuals are with using technology to find, evaluate, and apply health information. The questionnaire consists of 8 items, each scored on a 5-point Likert scale. The total score is calculated by summing the responses from all 8 questions, resulting in a score that ranges from 8 to 40. Higher scores indicate a greater perceived level of eHealth literacy.

**BADH.** The questionnaire evaluates attitudes concerning key aspects of bioethics such as data privacy, patient autonomy, and trust in digital health. The validated version consists of eight items organized into two dimensions and utilizes a 5-point Likert scale for responses. The total score is obtained by summing the responses to all eight questions, with possible scores ranging from 0 to 32. Higher scores indicate more positive attitudes.

## Statistical analysis

After conducting factor analysis, the original structure of the BADH was revised, resulting in the emergence of two distinct domains. For each domain of the BADH, a score was calculated by summing the individual item scores. The first domain

**Table 1. Patients socio-demographics.**

| | S-2<br>n = 100 | S-3<br>n = 120 |
|---|---|---|
| Age, years | 50 (37.3-63) | 47.5 (33-60.8) |
| Females* | 67 (67) | 77 (64.2) |
| Years of scholarship | 12 (12-17) | 12 (12-17) |
| Living with other people* | 95 (95) | 110 (91.7) |
| 15-40 years old | 60 (63.2) | 69 (62.7) |
| 41-60 years old | 22 (23.2) | 21 (19.1) |
| ≥ 61 years old | 62 (65.3) | 78 (70.9) |
| Number of the people they live with¹ | 2 (1-4) | 2.5 (1-4) |
| Occupation* | | |
| *Formal and non-formal job* | 41 (41) | 59 (49.2) |
| *Unemployed* | 13 (13) | 17 (14.2) |
| *Housewife* | 28 (28) | 27 (22.5) |
| *Student* | 6 (6) | 7 (5.8) |
| *Retired* | 12 (12) | 10 (8.3) |
| Patients benefiting from a regular monthly income*<br>Monthly income ≤ 500 USD¹* | 61 (61)<br>40 (65.5) | 77 (64.2)<br>49 (63.6) |

*Data presented as median (IQR) or otherwise indicated. *Number (%) of patients. ¹Among those with the characteristic.*

had a possible score range of 0–12, while the second domain had a possible score range of 0–20. Additionally, an overall BADH score was computed by adding the individual item scores from the domains. The final score could range from 0 to 32, with higher scores indicating a more favorable attitude.

Descriptive statistics were performed to describe the variables of the patients included in the three samples, with frequencies and percentages for categorical variables or the mean/median and standard deviation (SD)/Q25-Q75 for continuous variables with normal/non-normal distribution.

Face and content validity were examined with agreement percentages. Lawshe/Tristan's content validity ratio was calculated for the BADH (mean of individuals' content validity ratios) [38]. Construct validity was assessed using exploratory factor analysis (principal components), followed by confirmatory factor analysis [37]. Cronbach's α was used to assess the internal consistency of the questionnaire. For temporal stability/test-retest, intra-class correlation coefficients (ICC) and their 95% confidence intervals (CI) were calculated using a single measurement, absolute-agreement, 2-way mixed-effects model. Cronbach's α, ICC, and 95% CI interpretations followed published recommendations [39]. Finally, the floor and ceiling effects of the questionnaire were determined as the percentage of patients who achieved the lowest and highest score on the scale, respectively.

A multiple linear regression analysis was used to investigate the relationship between the BADH and the eHEALS scores. The following variables were considered as covariates: age, sex, occupation, access to internet, access to any digital device, use of a social media network and genital skin localization. Numerous cross-sectional studies have identified various factors that influence eHealth literacy, and their findings have been summarized in a recent systematic review [40]. This review highlighted key factors affecting eHealth literacy across three dimensions: actions (such as internet usage), social determinants (including age, ethnicity, income, employment status, education, perceived usefulness, and self-efficacy), and health status (related to specific diseases). Additionally, we included genital skin location as a covariate, due to the known stigmatization associated with skin disorders [41]. Finally, variable interaction effects were also examined. To evaluate the adequacy of the linear regression model, we assessed several key assumptions: linearity, homoscedasticity, normality of residuals, and the absence of multicollinearity. We analyzed the residual Q-Q plots, created scatterplots of each predictor against the residuals, conducted the Shapiro-Wilk test on the residuals, and calculated the variance inflation factors for all predictors. Additionally, we assessed the overall model fit and the effect sizes of individual predictors using the adjusted $R^2$.

In order to conduct an inferential analysis of DHL, we categorized patients into two groups: high DHL and low DHL. This classification was based on a statistical criterion using the 75th percentile of the data distribution as the cutoff point. We performed a bivariate comparison of all characteristics among patients based on their levels of DHL, using appropriate tests. The Mann-Whitney U test was used to compare continuous variables when the data did not show a normal distribution (Kolmogorov-Smirnov). Fisher's exact test or $X^2$ test was used to compare proportions. Additionally, the analysis was repeated using the median value of the data distribution as the cutoff point for high/low DHL, as previously reported [42,43].

To define factors associated with high DHL as a binary dependent variable, we performed multiple logistic regression analysis. We initially conceived a global model where variables' inclusion was based on their statistical significance in the univariate analysis ($p \leq 0.10$). To evaluate the adequacy of the logistic regression model, we assessed the linearity of the log-odds and checked for multicollinearity. This involved examining scatter plots of the explanatory variables against their natural logarithms and calculating the Spearman correlation coefficients between the explanatory variables. Additionally, we assessed the model fit and effect sizes using the Hosmer–Lemeshow test and the Nagelkerke pseudo-$R^2$.

Results are expressed as adjusted Odds Ratios (exponentiated regression coefficients, exp[$\beta$]) and their 95% CI.

Missing data were below 1%, and no imputation was performed.

All statistical analyses were performed using Statistical Package for the Social Sciences version 21.0 (SPSS, Chicago, IL) except the confirmatory factor analysis performed in the package Jeffrey's Amazing Statistics Program version 0.19.3 (JASP, University of Amsterdam). A value of $p < 0.05$ was considered statistically significant.

# Results

## Samples' description

There were 46 patients included in S-1, 100 in S-2, and 120 in S-3.

Tables 1-3 summarize the socio-demographics of the patients (Table 1), factors related to patients' digital access and usage (Table 2), and the factors related to the patients' dermatological diagnoses (Table 3) across S-2 and S-3. We do not have complete data from the patients included in pilot testing (S-1).

The patients were predominantly middle-aged females with an average of 12 years of formal education. Most lived with others, typically individuals aged 61 years or older, as well as those in the 15–40-year age range. Additionally, the majority of participants were employed and received a regular monthly income, although most earned less than the equivalent of 500 USD per month.

Most patients had full access to paid internet, although a smaller percentage rated the quality of their internet connectivity as good. Additionally, the majority of participants owned a mobile phone with internet access, whereas only a minority had access to a computer, tablet, or smartwatch. Finally, most patients used email, and the most frequently used social media platforms were WhatsApp, Facebook, and YouTube.

The most common reasons for dermatological consultations were dry skin, skin irritation, and sunspots in S-3; meanwhile in S-2, they were sunspots, nail changes, and skin irritation. Skin lesions were found in the genital area in a lower

**Table 2. Patients' digital access and usage characteristics.**

| | S-2<br>n = 100 | S-3<br>n = 120 |
|---|---|---|
| Access to internet | 90 (90) | 106 (88.3) |
| Access to paid internet[1] | 84 (93.3) | 102 (96.2) |
| Full (every day) access to internet[1] | 85 (94.4) | 99 (93.4) |
| Good quality of internet connectivity | 59 (65.6) | 62 (58.5) |
| Availability of a mobile phone with internet access | 98 (98) | 118 (98.3) |
| Own mobile phone[1] | 90 (91.8) | 115 (97.5) |
| Availability of a computer | 44 (44) | 56 (46.7) |
| Own computer[1] | 30 (68.2) | 44 (78.6) |
| Availability of a tablet | 16 (16) | 11 (9.2) |
| Own tablet[1] | 11 (68.8) | 10 (90.9) |
| Availability of a smart watch | 18 (18) | 10 (8.3) |
| Use of email | 83 (83) | 98 (81.7) |
| Use of social media | | |
| Whatsapp | 94 (94) | 112 (93.3) |
| Telegram | 20 (20) | 20 (16.7) |
| Facebook | 69 (69) | 87 (72.5) |
| X (formerly Twitter) | 22 (22) | 18 (15) |
| Youtube | 71 (71) | 83 (69.2) |
| Instagram | 44 (44) | 48 (40) |
| TikTok | 30 (30) | 42 (35) |
| LinkedIn | 11 (11) | 12 (10) |
| Other | 0 | 0 |

*Data presented as Number (%) of patients. [1]Among those with the characteristic.*

**Table 3. Factors related to the patients' dermatological diagnoses.**

| | S-2 n = 100 | S-3 n = 120 |
|---|---|---|
| **Reason for dermatological consultation** | | |
| Dry skin | 11 (11) | 40 (33.3) |
| Drug reaction | 5 (5) | 6 (5) |
| sunspots | 18 (18) | 24 (20) |
| Skin irritation/Allergies | 12 (12) | 28 (23.3) |
| Skin cancer | 11 (11) | 11 (9.2) |
| Nails changes | 14 (14) | 12 (10) |
| Urticaria (hives) | 10 (10) | 15 (12.5) |
| Acne | 9 (9) | 10 (8.3) |
| Mole revision | 4 (4) | 14 (11.7) |
| Other | 43 (43) | 44 (36.7) |
| **Lesions localization** | | |
| Face | 31 (31) | 74 (61.7) |
| Genitals | 5 (5) | 17 (14.2) |
| Other areas of the body | 69 (69) | 80 (66.7) |
| **Comorbidities** | | |
| Diabetes | 15 (15) | 22 (18.3) |
| Hypertension | 24 (24) | 29 (24.2) |
| Thyroid disorders | 19 (19) | 32 (26.7) |
| Cardiac disorders | 4 (4) | 7 (5.8) |
| Neurological conditions | 4 (4) | 12 (10) |
| Rheumatological conditions | 27 (27) | 28 (23.3) |
| Hematological conditions | 6 (6) | 7 (5.8) |
| Cancer | 8 (8) | 18 (15) |
| Transplant | 5 (5) | 17 (14.2) |
| HIV infection | 3 (3) | 12 (10) |
| Other | 20 (20) | 32 (26.7) |

*Data presented as Number (%) of patients. ¹Among those with the characteristic.*

percentage of participants. Comorbid conditions were also common, particularly thyroid disorders, hypertension, and rheumatological conditions.

**Phase-1 results.** *BADH content and face validity:* Experts agreed on the items and scale response evaluation (100% agreement), face validity (80% agreement), and instructions clarity (100% agreement). The (mean) content validity ratio for the BADH was 0.94. During pilot testing, patients agreed on items and instructions clarity (89% and 95.7% agreement, respectively) and 91% on face validity. Patients' and experts' proposals were all adopted in the final BADH version used for the validation process.

*BADH construct validity:* The structure of the BADH underwent modifications after exploratory factor analysis, as shown in Fig 1. The 8 items were distributed into two dimensions instead of three, named as follows: "Trust" (Dimension I) and "Privacy" (Dimension II). The KMO measure of 0.691 and significant result ($X^2 = 233.354$, $p \leq 0.001$) for the Bartlett sphericity test confirmed the adequacy of the sample. A 2-factor structure was extracted, which accounted for 59.8% of the total variance. All factors had eigenvalues >1. The factors were equivalent to the dimensions.

A two-factor structure was maintained in the confirmatory factor analysis, and the internal consistency of the items was found to be adequate. The model explained 76.2% of the total variance, with a Root Mean Square Error of Approximation

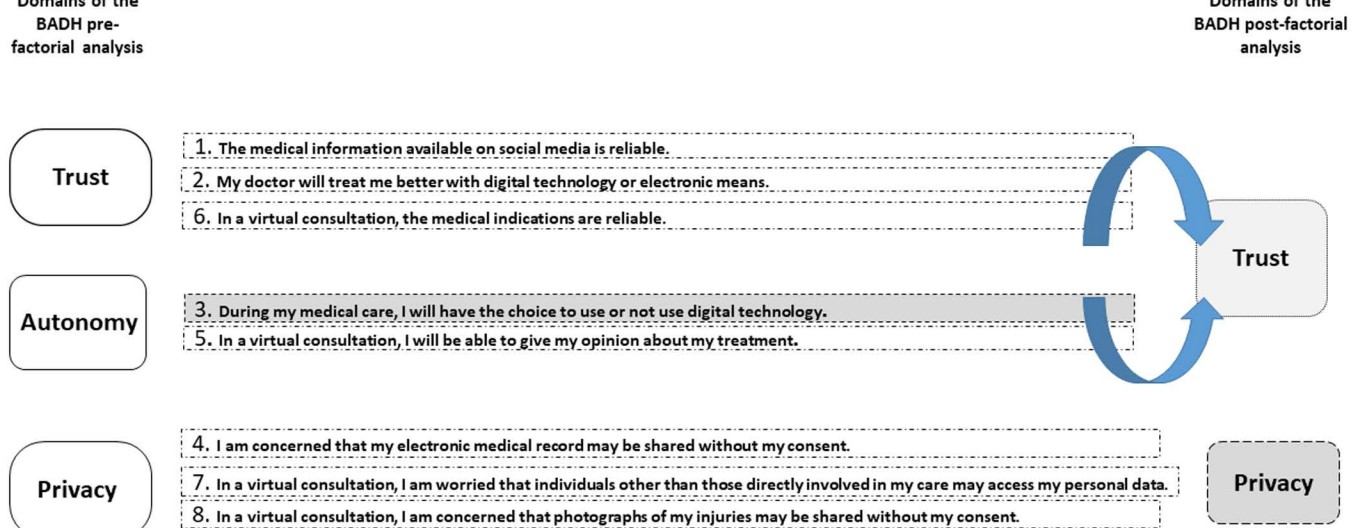

**Fig 1. BADH structure before and after factorial analysis.**

of 0.015 and a $X^2$ value of 19.487 (p = 0.426). Additionally, the KMO measure was 0.746, and the results of Bartlett's test of sphericity were significant ($X^2 = 318.438$, p ≤ 0.001), confirming the adequacy of the sample.

*BADH internal consistency and reliability:* Results of internal consistency (Cronbach's α) and reliability/test-retest (ICC and 95% CI) of the BADH and each dimension are presented in Table 4, which additionally presents floor and ceiling effects. The mean (±SD) of the time between the two measurements in the test-retest analysis was 7.04 ± 0.6 days. Test-retest was performed in 51 patients.

*BADH feasibility:* The majority of patients (97.8%) found the questionnaire format satisfactory. They felt that the time needed to complete the questionnaire was reasonable and expressed a willingness to participate. On average, patients took 9.4 minutes to complete it. After conducting pilot testing, it was found that patients benefited from assistance when the BADH was administered.

**Phase-2 results.** *Association between DHL and patients' attitudes regarding ethical aspects of digital health (primary objective):* In S-3, the median score for eHEALS was 26 (19–32), while the median score for BADH was 18 (14–22).

Results from multiple linear regression analysis are summarized in Fig 2 and highlight a positive association between eHEALS score and the BADH score, which was maintained with the trust dimension of the BADH but not with the privacy dimension. The interactions between variables did not influence the observed results.

*Factors associated with DHL:* The eHEALS cut-off value for classifying individuals with high DHL was set at 32 or above, representing the 75th percentile of the data distribution. Among the patients studied, 89 individuals (74.2%) were found to have low DHL. S1-3 Tables (Please see S1, S2 and S3 Table) provide a summary comparing the

**Table 4. BADH internal consistency, reliability/temporal stability, and floor and ceiling effects.**

|  | Cronbach´s α | ICC (95% CI)* | Floor/ceiling effect (%) |
|---|---|---|---|
| BADH | 0.686 | 0.684 (0.581-0.770) | 0/0 |
| Dimension-1 (Trust) | 0.772 | 0.772 (0.692-0.835) | 3/4 |
| Dimension-2 (Privacy) | 0.796 | 0.796 (0.715-0.857) | 13/12 |

ICC=Intraclass Correlation coefficient.CI=Confidence Interval. *Limited to 51 patients.

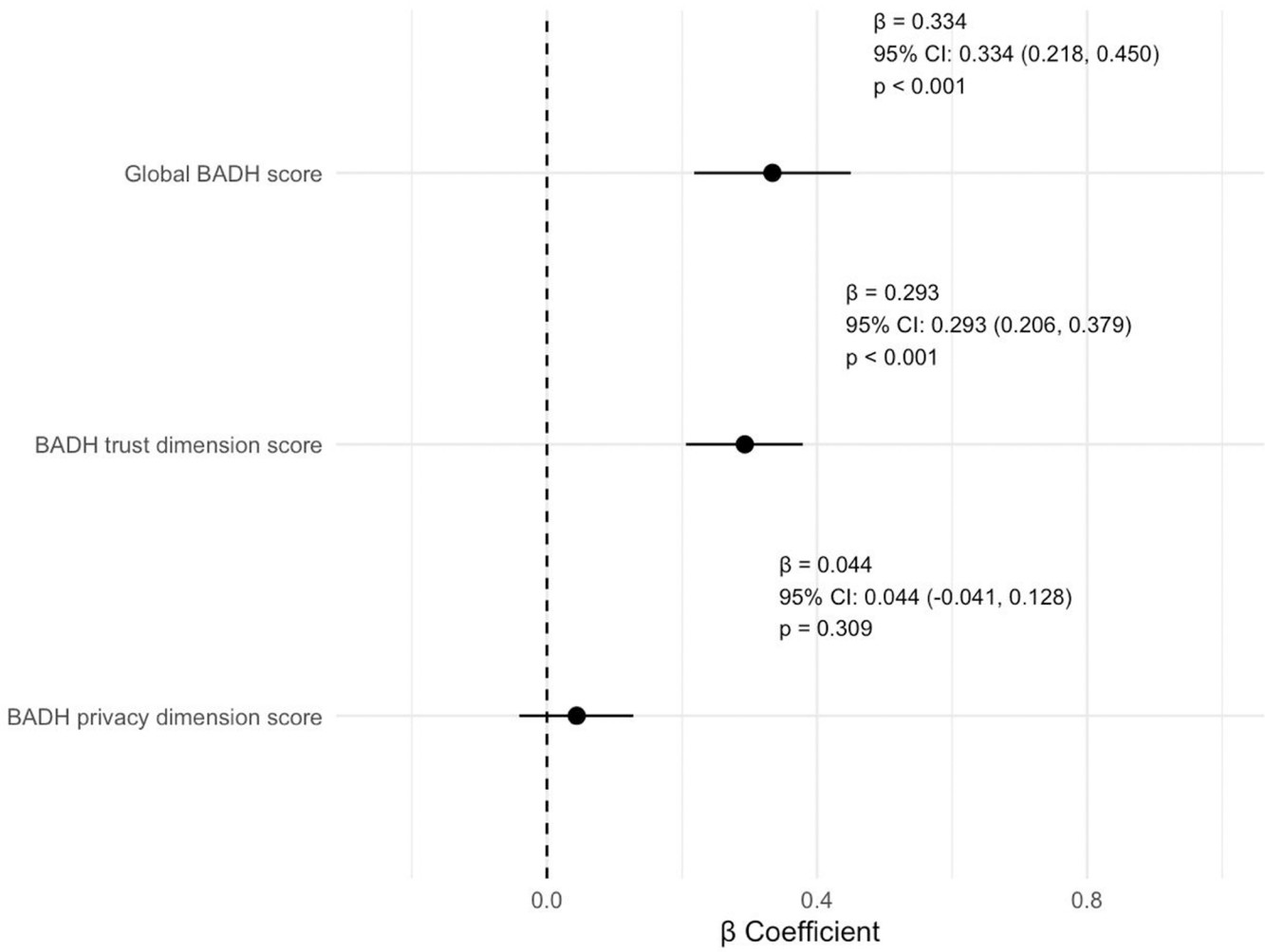

**Fig 2. Relationship between the BADH and the eHEALS scores (Multiple linear regression analysis).** Forest plot showing the relationship between Bioethical Attitudes towards Digital Health (BADH) and eHEALTH Literacy Scale (eHEALS) scores. The Figure displays standardized coefficients (β) with 95% confidence intervals (CI), and p-value. The following covariates were included in the model: age, sex, occupation, internet access, digital device access, social media use, genital skin localization. Data visualization was created with R-project software, version: 4.5.0.

socio-demographics (S1 Table), digital access and usage characteristics (S2 Table), and factors related to the patients' dermatological diagnoses (S3 Table) between those with high DHL and their counterparts.

In summary, patients with high DHL levels were more educated, reported having access to a tablet, and sought dermatological consultations primarily due to drug reactions. Additionally, these patients had a higher prevalence of cardiac disorders as comorbidities compared to their counterparts. They were also more likely to own a computer and use email, and they had other comorbidities as well.

The following variables were included in the logistic regression analysis: years of education, computer ownership, availability of a tablet, email use, drug reactions as the reason for seeking dermatological consultation, and the presence of cardiac disorders along with other comorbidities.

Years of education, presence of other comorbidities, and availability of tablets were associated with high DHL (Pseudo $R^2 = 0.294$) (Table 5).

**Table 5. Results from multiple logistic regression analysis to identify variables associated with high DHL, based on two different eHeals score cutoffs.**

| Factors/Dimensions[1] *Variables* | High-DHL (eHEALS score >32) | High-DHL (eHEALS score>26) |
|---|---|---|
| **Actions** | | |
| *Availability of tablets* | 4.289, 0.973–18.899, p=0.054 | |
| **Social determinants** | | |
| *Years of education* | 1.329, 1.040–1.698, p=0.023 | 1.171, 1.048–1.308, p=0.005 |
| *Years of age* | | 0.958, 0.932–0.983, p<0.001 |
| **Health status** | | |
| *Presence of other comorbidities* | 7.238, 1.586–33.023, p=0.011 | |
| *Cardiac comorbidity* | | 17.029, 1.424–203.714, p=0.052 |

[1]*Hua Z, Yuqing S, Qianwen L, Hong C. Factors Influencing eHealth Literacy Worldwide: Systematic Review and Meta-Analysis.J Med Internet Res 2025;27:e50313.* https://doi.org/10.2196/50313.

*Data are presented as ß exp, 95% CI and p value*

When the median value of the data distribution was used as the cutoff point to define high levels of DHL (median=26, N=61 [50.8%]), the analysis revealed similar associations (Pseudo $R^2$=0.299 (Table 5).

## Discussion

In this study, we assert that there is a positive association between patients' DHL and their attitudes toward specific ethical aspects of digital health. This association is particularly strong in terms of trust in digital health, while it does not extend to privacy concerns. Furthermore, we confirm that the BADH questionnaire, designed to assess patients' attitudes, demonstrates adequate psychometric properties. We also identify several risk factors that contribute to higher DHL. These factors can be categorized into three relevant dimensions based on a recent systematic review and meta-analysis: better education, which is part of the social determinants dimension; the presence of comorbid conditions, associated with the health status dimension; and access to digital technology, such as tablets, which falls under the actions dimension.

The study was conducted at a dermatology service within an urban academic center. Academic centers provide several advantages for clinical research, including multidisciplinary collaboration, a sense of community, a focus on societal issues, and the promotion of ongoing learning [44]. DHL of participants was assessed using a Spanish version of the eHEALS, which has demonstrated adequate psychometric properties for evaluating this construct in Spanish-speaking populations [10,13,14]. Patients' attitudes towards key bioethics domains related to digital health were measured using the BADH questionnaire, which was specifically designed and previously validated for this purpose. This validation process was essential, as the current study involves complex theoretical constructs that must be operationalized into measurable variables using appropriate instruments to minimize bias and avoid misclassification of patients.

Our primary objective examined the relationship between DHL and attitudes towards the ethical aspects of digital health.

In recent years, the rising prevalence of chronic diseases has placed a significant burden on healthcare systems. The implementation of digital health modalities has been effectively integrated into disease management, demonstrating promising results regarding cost-effectiveness [45–48] and the observation of the ethical aspects of access to healthcare [49]. However, criticisms have emerged, arguing that empirical evidence supporting the beneficial impact of most eHealth technologies is often lacking or, at best, modest [50,51]. While the absence of evidence does not equate to proof of ineffectiveness, reports of negative consequences highlight the importance of evaluating potential risks—both anticipated and otherwise [50]. Meanwhile, the benefits of digital health have extended to the field of dermatology [52], where healthcare professionals and patients have acknowledged the advantages of eHealth services in daily dermatology practice [29].

We focused on attitudes toward the moral aspects of digital health, utilizing the MFT framework. A key strength of this framework is its ability to examine people's underlying moral intuitions, which consistently align with their attitudes and behaviors. MFT posits that when individuals recognize a moral foundation for their attitudes, those attitudes are more predictive of behavior and are less likely to change [53,28]. This connection has been observed across various clinical contexts, particularly during the COVID-19 pandemic [28,54]. Applying this framework to the current study was not meant to dictate the BADH content. Instead, it supports the notion that positive moral attitudes toward digital health may facilitate its acceptance. This perspective is based on the understanding that eHealth services are valuable due to their significant potential to enhance the health and well-being of both populations and individuals [55].

The observed association between DHL and the ethical aspect of trust in digital health—though not in relation to privacy concerns—suggests that DHL serves as a new determinant of health [10,56]. Some authors even contend that DHL is a "super social determinant of health" due to its significant implications for the broader social determinants of health [56,57]. A recent literature review [10] has found that DHL is linked to favorable health outcomes, including health-promoting behaviors (such as health responsibility, stress management, exercise habits, self-realization, and social support), improved quality of life, enhanced mental and psychological states (which involve managing negative emotions, meta-cognition, and psychological well-being), as well as better disease control in patients with type 2 diabetes [58]. The current study further adds the construct of trust to this list. Trust, as a sociological construct, refers to people's expectations of goodwill, advocacy, and competence [59]. It manifests in different forms: Expectant trust arises from a patient's predisposition during a first encounter; experienced trust develops through familiarity over time; and identification-based trust is based on shared values [60]. Research indicates that trust is vital in healthcare. It enhances patient-physician communication [61], optimizes the time of both parties [59], and increases the effectiveness of prescribed treatments [61]. Patients who trust their doctors rate their care more favorably and spontaneously emphasize the importance of trust [62–67]. Moreover, a lack of trust in unfamiliar healthcare providers—often influenced by trust in organizations—can lead patients to avoid essential care, particularly detrimental for chronic disease patient biographies and contrary to medical principles [68]. From a public health perspective, fostering and maintaining trust in healthcare systems is essential for achieving better outcomes, especially during crises [69].

Finally, the lack of association between DHL and privacy concerns may be attributed to the well-known inherent privacy challenges associated with digital technologies [70–72], which has been confirmed among patients with dermatologic conditions [29].

The BADH questionnaire exhibited the following psychometric properties. Its validation involved thorough assessments of validity and reliability, with face and content validity determined by a panel of experts, including patients. While clinicians typically focus on biomedical factors, patient insights are crucial for understanding subjective experiences [73]. Reliability was measured through internal consistency and test-retest methods, following established guidelines [31]. The Cronbach's α values for the dimensions demonstrated acceptable reliability, with the overall BADH score nearing 0.7 [74]. The ICC values indicated moderate reliability for the total scale and both dimensions [75]. Construct validity was confirmed by the KMO sampling and Bartlett's test of sphericity, which indicated the sample size was sufficient for exploratory factorial analysis [76]; furthermore, the BADH structure was subsequently validated through confirmatory factor analysis. Importantly, the questionnaire did not exhibit floor or ceiling effects, which can occur when over 15% of respondents score at the extreme ends, potentially masking significant changes [31]. During pilot testing, patients assessed the BADH as practical and suitable for individuals with low literacy levels. Lastly, factor structure changed after factorial analysis. A two-factor structure was determined to be appropriate and explained 59.8% of the total variance in exploratory factor analysis. This variation in structure may be attributed to the complexity of the trust construct and the different paths in which it might manifest, including participants' expectations regarding their right to exercise the autonomy principle [60].

Considering the current status of DHL as a new determinant of health (similar to socioeconomic level, income, education, age, race, ethnicity, and gender) we sought to explore the factors associated with high DHL and identified education, the presence of comorbid conditions, and access to digital technology, such as tablets.

Previous studies have established a link between overall education and DHL, which has been summarized in a recent systematic review and meta-analysis [40]. Guo et al. [77], in a random cohort of adults in Hong Kong, examined socioeconomic disparities in seeking web-based information on COVID-19 and eHealth literacy, and their associations with personal preventive behaviors during the COVID-19 pandemic. They observed a negative association between DHL and age, as well as a positive association between education and a higher level of mobile eHealth literacy. Interestingly, we also confirmed this negative association (age and DHL) when a high DHL was defined based on an eHEALS score greater than 26. Additionally, a similar positive association was found among a sample of university students, where educational level was the primary factor influencing differences in responses to digital health literacy [78].

Comorbid conditions might be considered a surrogate for a higher number of consultations, some of which might be offered in the modality of eHealth and eventually serve as training. Education and training have been reported as successful interventions to improve DHL both in children and adults [10]. Meanwhile, lack of education and training has been noted among the most frequent barriers to digital health [40,79–87], and these results have been replicated among patients with dermatological conditions [29]. There are several reasons for digital exclusion, and insufficient skills, suitable to be improved with repeated exposure, is one of them [79].

A similar argument can be applied to the observed association between access to digital technology, such as a tablet, and higher DHL. Few patients had a tablet available compared to those with access to mobiles, and this subpopulation might represent those with more possibilities of implementing health services and might favor a more positive attitude toward digital health services. Positive attitudes towards the use of digital health services has been observed among professionals with more experience in using health applications [28], and this finding could extend to patients. The availability of tablets may promote their use and can be considered a variable linked to one of the three dimensions ("actions") influencing eHealth literacy [39].

This research has several limitations that should be noted. First, participants were recruited from a single urban academic center, likely resulting in a sample with more severe chronic disease phenotypes and greater exposure to digital health, which may have influenced their attitudes toward ethical aspects of digital health and affected external validity. Second, the Cronbach's alpha for the BADH questionnaire was below (but close to) 0.7, raising questions about reliability, possibly due to a limited number of items included in the questionnaire and the complex nature of the construct evaluated [74]. We also did not examine criterion validity for the BADH, and a sample size of 100 patients might be considered insufficient for factorial analysis [88]. Third, the standardized order of the questionnaires´ administration could have impacted participant compliance and attention. Fourth, we investigated a limited number of factors associated with high DHL, defined according to a specific statistical criterion. When we repeated the analysis using a different statistical criterion, we obtained similar results, which supports the clinical relevance of our findings. However, categorizing continuous variables can lead to misleading interactions [89]. Also, while the median of a data distribution is commonly used, in certain medical and epidemiological contexts, it can be more appropriate to categorize at extreme values as we did [89]. Fifth, the findings may be specific to patients from LATAM, considering the significant role of economic and cultural contexts in shaping DHL and attitudes toward ethical aspects of digital health. Finally, participant selection relied on convenience sampling, which may overlook patients who do not attend clinics or follow-ups due to challenges with the eHealth tools that could be used in the dermatology outpatient clinic.

## Conclusions

DHL is connected to personal moral views on digital health, particularly in terms of trust. The initial evaluation of the psychometric properties of the BADH questionnaire demonstrated adequate validity but revealed limitations in its internal consistency. This highlights the necessity for further research to address these issues and enhance the reliability of the questionnaire. Moreover, factors such as higher education, the presence of comorbid conditions, and access to digital technology, such as tablets, are associated with increased DHL. To our knowledge, the BADH questionnaire is

the first tool designed to assess patients' attitudes toward the ethical aspects of digital health in the context of cutaneous conditions. However, it was developed using a specific group of patients, and the findings need to be confirmed in other populations.

## Supporting information

**S1 Appendix. Bioethical attitudes regarding digital health questionnaire (BADH). Spanish and English versions.**
(PDF)

**S1 Table. Comparison of socio-demographic characteristics between participants with high DHL levels and their counterparts.**
(PDF)

**S2 Table. Comparison of patients' digital access and usage characteristics between participants with high DHL levels and their counterparts.**
(PDF)

**S3 Table. Comparison of factors related to the patients' dermatological diagnoses between participants with high DHL levels and their counterparts.**
(PDF)

## Author contributions

**Conceptualization:** Ana Lilia Ruelas-Villavicencio, Irazú Contreras-Yáñez, Virginia Pascual-Ramos.

**Data curation:** Roxana Paola Gómez-Ruiz, María Clara Zagaglia Del Valle, Andrea Malagón-Liceaga.

**Formal analysis:** Ana Lilia Ruelas-Villavicencio, Irazú Contreras-Yáñez, María Clara Zagaglia Del Valle, Andrea Malagón-Liceaga, Virginia Pascual-Ramos.

**Investigation:** Ana Lilia Ruelas-Villavicencio, Roxana Paola Gómez-Ruiz, María Clara Zagaglia Del Valle, Andrea Malagón-Liceaga.

**Methodology:** Irazú Contreras-Yáñez, Virginia Pascual-Ramos.

**Supervision:** Ana Lilia Ruelas-Villavicencio.

**Validation:** Ana Lilia Ruelas-Villavicencio, Irazú Contreras-Yáñez, Roxana Paola Gómez-Ruiz, María Clara Zagaglia Del Valle, Andrea Malagón-Liceaga, Virginia Pascual-Ramos.

**Visualization:** Ana Lilia Ruelas-Villavicencio, Irazú Contreras-Yáñez, Virginia Pascual-Ramos.

**Writing – original draft:** Virginia Pascual-Ramos.

**Writing – review & editing:** Ana Lilia Ruelas-Villavicencio, Irazú Contreras-Yáñez, Roxana Paola Gómez-Ruiz, María Clara Zagaglia Del Valle, Andrea Malagón-Liceaga.

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
