## [Decision Letter · Decision Letter 0]

24 Jun 2025

Dear Dr. Pascual-Ramos,

Thank you for submitting your manuscript to PLOS ONE. After careful consideration, we feel that it has merit but does not fully meet PLOS ONE’s publication criteria as it currently stands. Therefore, we invite you to submit a revised version of the manuscript that addresses the points raised during the review process.

We look forward to receiving your revised manuscript.

Kind regards,

Blessing Onyinye Ukoha-kalu, B.Pharm, M.Pharm, Ph.D

Academic Editor

PLOS ONE

Journal Requirements:

2. You have indicated that data is available from [sergio.hernandezj@incmnsz.mx].  Please can we ask you to provide us with a general contact email address for the data requests, so readers can request access in perpetuity. If a general email is not available please provide a link to a website where readers can obtain access to data.

Reviewers' comments:

Reviewer's Responses to Questions

**Comments to the Author**

1. Is the manuscript technically sound, and do the data support the conclusions?

Reviewer #1: Yes

Reviewer #2: Partly

Reviewer #3: Yes

2. Has the statistical analysis been performed appropriately and rigorously?

Reviewer #1: I Don't Know

Reviewer #2: No

Reviewer #3: Yes

3. Have the authors made all data underlying the findings in their manuscript fully available?

Reviewer #1: Yes

Reviewer #2: No

Reviewer #3: No

4. Is the manuscript presented in an intelligible fashion and written in standard English?

Reviewer #1: Yes

Reviewer #2: Yes

Reviewer #3: Yes

Reviewer #1: Excellent piece! The abstract was clear and concise, effectively summarizing that the tables were well thought out and explained the data clearly. The article’s structure, coherence, and depth of analysis was superb. I would just like a graph please just to add some variation to the tables.

Reviewer #2: Review Comments to the Author

The manuscript presents a timely and relevant exploration of the relationship between digital health literacy (DHL) and attitudes toward the ethical dimensions of digital health, particularly within a dermatologic patient population. It is commendable that the authors aimed to validate a novel instrument (BADH) and also attempted to establish empirical associations using eHEALS scores.

However, several important concerns emerged during the review:

Technical Soundness and Support for Conclusions (Partly):

While the study is conceptually sound and well-structured, the robustness of the conclusions is partially limited by methodological gaps. The conclusions regarding the association between DHL and moral attitudes, especially trust, are aligned with the results. However, some claims (e.g., the broader implications of the BADH tool or its relevance beyond the specific setting) may overreach the data presented. A more cautious interpretation is advised

Statistical Analysis (No)

Although the manuscript reports descriptive and inferential statistics, the rigor and depth of the statistical analysis require improvement. For instance, multiple linear regression is used but the rationale for including specific covariates is not fully justified, and interaction effects are not explored. Moreover, the criteria for grouping DHL (based on 75th percentile) may introduce bias and limit generalizability. There is also no clear discussion on model diagnostics, assumptions, or effect size interpretations.

Data Availability (No)

The data are not openly available, which limits transparency and reproducibility. While the authors justify this based on local data protection laws, PLOS policy expects data to be shared upon publication, with only rare and specific exceptions. The current data access route (via IRB approval) presents a significant restriction.

Language and Presentation (Yes)

The manuscript is well-written in standard academic English. The content is intelligible, with minor grammatical or typographical issues. The writing is coherent, and the study flow is easy to follow.

Reviewer #3: The manuscript is generally well-structured and addresses an important gap in digital health and ethics research, particularly in underrepresented populations such as this. The overall methodology is clearly outlined, and the statistical techniques employed, such as exploratory factor analysis, Cronbach’s alpha, and multiple regression are appropriate for the study’s aims. However, there are some technical concerns that impact the strength of the conclusions.

The validation of the BADH questionnaire, while a valuable contribution, would benefit from a larger and more diverse sample size to ensure stability of the factor structure and internal consistency. The Cronbach's alpha for the full scale (0.686) falls just below the accepted threshold for reliability, and the study lacks confirmatory factor analysis to reinforce the two-dimensional structure identified. Additionally, the use of an arbitrary cutoff at the 75th percentile to define "high" DHL raises concerns about the robustness of some subgroup analyses.

Despite these limitations, the association found between digital health literacy and ethical attitudes, especially regarding trust is supported by the data and aligns with existing literature. The conclusions drawn are cautious and largely appropriate, but should more clearly acknowledge the psychometric limitations and sampling constraints that may affect generalizability.

Data Availability:

The authors have provided a detailed data availability statement, explaining that the complete dataset cannot be made openly accessible due to concerns about patient identification and compliance with Mexican data protection laws. They emphasize that individual-level data are protected under national regulations and participant consent agreements.

While the dataset is not publicly available, the authors offer a clear process for requesting access through their local Institutional Review Board (IRB), and provide contact information for the IRB chair. This approach respects ethical and legal obligations and is acceptable under PLOS ONE’s data policy, which allows for restricted access in justified cases involving sensitive data.

In summary, although the data are not fully open, the authors have made a reasonable and transparent effort to ensure access under appropriate oversight.

Some comments

Clarifying conceptual framing: The manuscript references Moral Foundations Theory but doesn't adequately explain how the BADH instrument operationalizes its principles, especially since only “trust” and “privacy” dimensions are measured.

Addressing statistical inconsistencies: There are mismatches between reported beta coefficients and confidence intervals, and the definition of “high DHL” using the 75th percentile lacks theoretical or empirical justification.

**Do you want your identity to be public for this peer review?** For information about this choice, including consent withdrawal, please see our Privacy Policy

Reviewer #1: **Yes: ** Nabeela Adam

Reviewer #2: **Yes: ** Bruce Ayabilla Abugri

Reviewer #3: No

---

## [Author Response · Author response to Decision Letter 1]

9 Jul 2025

JOURNAL REQUIREMENTS:

Response. We have ensured that the manuscript meets the journal's requirements, including the names of the attached files; therefore, we have modified the file names of the figures and supplementary material accordingly.

2. You have indicated that data is available from [sergio.hernandezj@incmnsz.mx]. Please can we ask you to provide us with a general contact email address for the data requests, so readers can request access in perpetuity. If a general email is not available please provide a link to a website where readers can obtain access to data.

Response. We have modified the data availability statement to provide a general contact email address for data requests, allowing readers to request access on an ongoing basis.

RESPONSES TO REVIEWERS:

Reviewer #1: Excellent piece! The abstract was clear and concise, effectively summarizing that the tables were well thought out and explained the data clearly. The article’s structure, coherence, and depth of analysis was superb. I would just like a graph please just to add some variation to the tables.

Maybe add a bar chart or line graph to add some variation to the tables. Well done to the authors.

Response. We sincerely value the reviewer’s insightful feedback. In response, we have removed Table 5 and transformed it into figure 2.

Reviewer #2: Review Comments to the Author

The manuscript presents a timely and relevant exploration of the relationship between digital health literacy (DHL) and attitudes toward the ethical dimensions of digital health, particularly within a dermatologic patient population. It is commendable that the authors aimed to validate a novel instrument (BADH) and also attempted to establish empirical associations using eHEALS scores.

Response. We appreciate the reviewer's feedback.

However, several important concerns emerged during the review:

Technical Soundness and Support for Conclusions (Partly):

While the study is conceptually sound and well-structured, the robustness of the conclusions is partially limited by methodological gaps. The conclusions regarding the association between DHL and moral attitudes, especially trust, are aligned with the results. However, some claims (e.g., the broader implications of the BADH tool or its relevance beyond the specific setting) may overreach the data presented. A more cautious interpretation is advised.

Response. We have been more cautious in the conclusions section.

Statistical Analysis (No)

Although the manuscript reports descriptive and inferential statistics, the rigor and depth of the statistical analysis require improvement. For instance, multiple linear regression is used but the rationale for including specific covariates is not fully justified, and interaction effects are not explored.

Response. We have updated the Statistical Analysis and Results section with the required information.

Moreover, the criteria for grouping DHL (based on 75th percentile) may introduce bias and limit generalizability

Response. We appreciate the reviewer's feedback and acknowledge it as a limitation of our study. In response, we have improved our initial proposal by using the median of the data distribution as the cutoff value for defining high DHL, as suggested by two authors. The results from this updated analysis have been discussed in the relevant section.

There is also no clear discussion on model diagnostics, assumptions, or effect size interpretations.

Response. We have updated the Statistical Analysis section with the required information.

Data Availability (No)

The data are not openly available, which limits transparency and reproducibility. While the authors justify this based on local data protection laws, PLOS policy expects data to be shared upon publication, with only rare and specific exceptions. The current data access route (via IRB approval) presents a significant restriction.

Response. The complete data supporting the findings of this study is not publicly accessible due to legal and ethical concerns regarding patient identification and the sensitivity of the information. However, we provide a clear and open pathway for accessing the data, as it is available to anyone who submits a reasonable request via email. As the reviewer noted, we are also concerned about research integrity and understand the importance of making data publicly available to uphold that integrity. However, we firmly believe that individual rights must take precedence.

Language and Presentation (Yes)

The manuscript is well-written in standard academic English. The content is intelligible, with minor grammatical or typographical issues. The writing is coherent, and the study flow is easy to follow.

Response. We appreciate the reviewer's feedback.

Additional comments:

Abstract; Clarification and Precision

Line 22–23: Define “digital health literacy (DHL)”; briefly. Readers unfamiliar with the term may benefit from a concise explanation.

Response. We have provided a definition.

Line 39: Instead of “p ≤ 0.0001”, revise to “p<0.001” to align with APA and biomedical reporting conventions.≤

Response. The data has been revised in the abstract and Figure 2, previously Table 5, based on your suggestions. Thank you for the recommendation to enhance the manuscript.

2. Introduction –Explicit Problem Statement and Context

Lines 105–106: The problem statement is implied but not clearly articulated. Add a focused paragraph stating the knowledge gap: e.g., “Despite growing attention to digital health tools, the ethical dimensions of patient engagement—particularly how DHL shapes moral attitudes remain understudied, especially in Latin American settings.”

Lines 108–109: Justify the focus on dermatologic patients more explicitly. Explain whether this population is particularly affected by digital health applications or ethical dilemmas in care.

Response: We have revised the introduction based on the reviewer's suggestions.

3. Methodology – Sampling and Analysis Transparency

Lines 147–148: Clarify how the use of convenience sampling may limit generalizability. Acknowledge potential bias in participant selection.

Response. We have acknowledged this as a limitation in the relevant section.

Line 236: When discussing regression modeling, explain how multicollinearity was handled

(e.g., via correlation matrix or VIF).

Response. We have revised the relevant section and incorporated the suggestion.

Lines 228–229: Justify the use of the 75th percentile as a cut-off for defining high DHL. Was this based on prior literature or purely exploratory?

Response. We have discussed that point and also included it as a limitation in the study. In response, we have improved our initial proposal by using the median of the data distribution as the cutoff value for defining high DHL, as suggested by two authors. The results from this updated analysis have been discussed in the relevant section.

4. Instrument Validation – Reliability Concerns

Line 292: The Cronbach’s alpha of 0.686 for the overall BADH is below the conventional threshold of 0.70.

Response. We have recognized this as a limitation and offered a possible explanation for it.

Reviewer #3: The manuscript is generally well-structured and addresses an important gap in digital health and ethics research, particularly in underrepresented populations such as this. The overall methodology is clearly outlined, and the statistical techniques employed, such as exploratory factor analysis, Cronbach’s alpha, and multiple regression are appropriate for the study’s aims. However, there are some technical concerns that impact the strength of the conclusions.

Response. We appreciate the reviewer's feedback.

The validation of the BADH questionnaire, while a valuable contribution, would benefit from a larger and more diverse sample size to ensure stability of the factor structure and internal consistency. The Cronbach's alpha for the full scale (0.686) falls just below the accepted threshold for reliability, and the study lacks confirmatory factor analysis to reinforce the two-dimensional structure identified. Additionally, the use of an arbitrary cutoff at the 75th percentile to define "high" DHL raises concerns about the robustness of some subgroup analyses.

Despite these limitations, the association found between digital health literacy and ethical attitudes, especially regarding trust is supported by the data and aligns with existing literature. The conclusions drawn are cautious and largely appropriate, but should more clearly acknowledge the psychometric limitations and sampling constraints that may affect generalizability.

Response. We appreciate the reviewer’s comments. In the limitations section, we acknowledge that the Cronbach's alpha value is below 0.7 and provide a possible explanation for this finding. We also recognize that the sample size may limit the factorial analysis. Furthermore, we note that the specific characteristics of the patients could affect the external validity of the results.

In the patients and methods section, we have discussed using the 75th percentile to establish the health literacy threshold. We also conducted a repeated analysis using the median value of the data distribution as an alternative cutoff for health literacy levels.

We have recognized as a limitation of the study that spurious associations may arise from the categorization of continuous variables.

Lastly, we have included a confirmatory factor analysis.

In our conclusions, we have adopted a more cautious approach by acknowledging some methodological limitations.

Data Availability:

The authors have provided a detailed data availability statement, explaining that the complete dataset cannot be made openly accessible due to concerns about patient identification and compliance with Mexican data protection laws. They emphasize that individual-level data are protected under national regulations and participant consent agreements.

While the dataset is not publicly available, the authors offer a clear process for requesting access through their local Institutional Review Board (IRB), and provide contact information for the IRB chair. This approach respects ethical and legal obligations and is acceptable under PLOS ONE’s data policy, which allows for restricted access in justified cases involving sensitive data.

In summary, although the data are not fully open, the authors have made a reasonable and transparent effort to ensure access under appropriate oversight.

Response. Thank you for your comment.

Some comments: Clarifying conceptual framing: The manuscript references Moral Foundations Theory but doesn't adequately explain how the BADH instrument operationalizes its principles, especially since only “trust” and “privacy” dimensions are measured.

Response. We propose an updated paragraph in the discussion section clarifying that key strength of MFT framework was its ability to examine people’s underlying moral intuitions, which consistently align with their attitudes and behaviors. Applying this framework to the current study was not meant to dictate the BADH content. Instead, it supports the notion that positive moral attitudes toward digital health may facilitate its acceptance.

Addressing statistical inconsistencies: There are mismatches between reported beta coefficients and confidence intervals, and the definition of “high DHL” using the 75th percentile lacks theoretical or empirical justification.

Response. We have updated the data. We have repeated analyses considering the median value of the data distribution, as previously done by two authors conveniently referred to. Additionally, we have added this as a limitation of the study.

---

## [Decision Letter · Decision Letter 1]

8 Aug 2025

Digital health literacy is linked to attitudes regarding the ethical aspects of digital health among patients with dermatologic comorbidities.

PONE-D-25-17804R1

Dear Dr. Pascual-Ramos,

We’re pleased to inform you that your manuscript has been judged scientifically suitable for publication and will be formally accepted for publication once it meets all outstanding technical requirements.

Kind regards,

Blessing Onyinye Ukoha-kalu, B.Pharm, M.Pharm, Ph.D

Academic Editor

PLOS ONE

Additional Editor Comments:

Thank you for taking the time to respond to reviewers' comments.

---

## [Editor Report · Acceptance letter]

PONE-D-25-17804R1

PLOS ONE

Dear Dr. Pascual-Ramos,

I'm pleased to inform you that your manuscript has been deemed suitable for publication in PLOS ONE. Congratulations! Your manuscript is now being handed over to our production team.

Kind regards,

on behalf of

Dr Blessing Onyinye Ukoha-kalu

Academic Editor

PLOS ONE